# Amaranth Oil Increases Total and LDL Cholesterol Levels without Influencing Early Markers of Atherosclerosis in an Overweight and Obese Population: A Randomized Double-Blind Cross-Over Study in Comparison with Rapeseed Oil Supplementation

**DOI:** 10.3390/nu11123069

**Published:** 2019-12-16

**Authors:** Monika Dus-Zuchowska, Jaroslaw Walkowiak, Anna Morawska, Patrycja Krzyzanowska-Jankowska, Anna Miskiewicz-Chotnicka, Juliusz Przyslawski, Aleksandra Lisowska

**Affiliations:** 1Department of Pediatric Gastroenterology and Metabolic Diseases, Poznan University of Medical Sciences, 60-572 Poznan, Poland; monzuchowska@gmail.com (M.D.-Z.); p.krzyzanowska81@gmail.com (P.K.-J.); chotnicka@ump.edu.pl (A.M.-C.); alisowska@ump.edu.pl (A.L.); 2Department of Bromatology, Poznan University of Medical Sciences, 60-354 Poznan, Poland; akm@ump.edu.pl (A.M.); jprzysla@ump.edu.pl (J.P.)

**Keywords:** atherosclerosis, amaranth oil, obesity

## Abstract

Background: Atherosclerosis (AT) is a chronic inflammatory process in which oxidative stress is the key event. Amaranth oil (AmO) has potential hypolipidemic and antiatherogenic effects. The aim of the study was to compare the effects of AmO and rapeseed oil (RaO) supplementation on expression of early markers of AT and lipid profile in obese or overweight subjects. Methods: A randomized, double-blinded cross-over study was conducted, in which participants took 20 mL of AmO in the first arm and 20 mL RaO in the second arm, switching after the washout period. Serum concentrations of adhesion molecules (sP-selectin, sVCAM-1), high-sensitivity C-reactive protein (hsCRP), asymmetric dimethylarginine (ADMA), and lipid profile were assessed before and after nutritional interventions. In addition, anthropometric parameters were measured. Results: The total (TC) and low-density lipoprotein (LDL) cholesterol concentrations increased significantly in the AmO group in comparison with RaO (ΔTC 5.52 ± 35 vs. −8.43 ± 17.65 mg/dL; *p* = 0.002 and 4.43 ± 34.96 vs. −7.55 ± 16.41 mg/dL; *p* = 0.002, respectively). There were no significant differences in other parameters analyzed between the groups. Conclusion: The use of AmO instead of RaO may increase cardiovascular risk in obese and overweight subjects.

## 1. Introduction

The epidemic of overweight and obesity with its wide spectrum of consequences is a major public health challenge of the 21st century. It affects more than a third of the global population and is associated with more mortality than being underweight [1,2]. Obesity is the paramount risk factor of atherosclerosis (AT), one of the most common health problems of modern civilization. There are three stages in the onset of atherogenesis: 1. adherence of mononuclear cells to endothelial cells; 2. migration of mononuclear cells into the subendothelial space; and 3. differentiation and transformation of mononuclear cells into macrophages and foam cells [3]. The first stage is triggered by adhesion molecules, such as vascular adhesion molecule-1 (VCAM-1), intracellular adhesion molecule-1 (ICAM-1), P-selectin, and E-selectin [3,4,5,6,7]. The second stage requires the presence of chemoattractants in the intima, mainly monocyte chemoattractant protein (MCP-1). In the third stage, infiltrated monocytes differentiate to macrophages, absorb modified lipoproteins, mainly oxidized low-density lipoprotein (LDL), finally differentiating into foam cells [8]. These events constitute the initial steps in the development of atherosclerosis (AT). These key stages in the pathogenesis of AT are inflammatory, preceded by oxidative stress in the endothelium of blood vessels [9,10,11]. Proven proatherogenic agents include high-sensitive C-reactive protein (hsCRP), mainly due to its proinflammatory action, and asymmetrical dimethylarginine (ADMA), which inhibits nitric oxide production by the endothelium [12,13]. The dysfunction (or activation) of the endothelium as a result of high serum levels of LDL and low levels of high-density lipoprotein (HDL) is related to obesity [14]. Thus, obese subjects are more at risk of plaque formation in their blood vessels, which could lead to potentially life-threatening cardiovascular events.

A longer life expectancy has prompted scientists to broaden their knowledge regarding the causes, mechanisms, and methods of AT prevention. The improvement of socio-economic conditions and public interest in so-called functional food has created the opportunity to introduce new, healthy eating habits that can minimize the risk of AT and obesity. A novel food ingredient which provides potential health benefits is the oil from the seeds of amaranth (*Amaranthus cruentus*). It contains valuable fatty acids, being composed of approximately 40% linoleic acid, 32% oleic acid, and 1% α-linolenic acid, as well as squalene, which has potential antioxidant, antihypertensive, and hypolipemic effects. Amaranth oil (AmO) contains ten times higher amount of squalene compared with olive oil. Moreover, AmO also contains derivatives of vitamin E (tocopherols and tocotrienols) and phytosterols, such as spinasterol, δ-7-stigmasterol, and δ-7-ergosterol [15].

The aim of the study was to compare the effects of AmO versus rapeseed oil (RaO) on novel markers of AT and the lipid profile in overweight and obese subjects. It was hypothesized that AmO would decrease the concentrations of proatherogenic agents, such as hsCRP, ADMA, adhesion molecules, and would improve lipid profile.

## 2. Materials and Methods

### 2.1. Patient Characteristics

The study enrolled 44 overweight or obese individuals, including 32 women and 12 men via written advertisements. The inclusion criteria were body mass index (BMI) ≥25, aged 18 years and older. The exclusion criteria involved a history of chronic systemic disease, celiac disease, liver disease, exocrine pancreatic insufficiency, current or recent (within preceding month) treatment with conjugated linoleic acid, statins, and current treatment with agents interfering with fat digestion/absorption (chitosan, orlistat, green tea). Written consent was obtained from all participants on entry into the study. Of the 51 registered subjects, 44 met the inclusion criteria of the study. A physical examination including the evaluation of body weight (BW), height, waist circumference (WC), BMI, and waist-to-hip ratio (WHR) was conducted four times: upon entry to the study, after first nutritional intervention, after washout period, and after second nutritional intervention. The study was performed in the Department of Pediatric Gastroenterology and Metabolic Diseases of Poznan University of Medical Sciences, Poland. Baseline characteristics of the study groups are presented in Table 1.

### 2.2. Study Design

The Project Was Designed as a Randomised, Double-Blind, Cross-Over Study, as Shown in the Flowchart in Figure 1.

On entry, all participants were randomly assigned to one of the arms. In arm I, the subjects were asked to administer 20 mL of AmO per day in the first nutritional intervention and 20 mL of RaO per day in the second nutritional intervention; in arm II, participants received 20 mL of RaO per day in the first nutritional intervention and 20 mL of AmO per day in the second nutritional intervention. The first and second nutritional interventions were separated by a three-week washout period. The investigators and participants were blinded from the computer-generated randomization codes until the end of all data analysis.

The AmO used in the study was extracted from *Amaranthus cruentus* and given in the form of Ol’Amar—dietary supplement produced by “Szarlat”, Lomza, Poland. The RaO used in the study was produced by VITACORN, Poznan, Poland. Both oils were stored in identical bottles. Neither patients nor research staff knew the content of the bottles. Doses of oil were applied in exchange for 20 g of fat used in the diet, so that the energy value of the diet did not change during the study. The basic characteristics of the diet are presented in Table 2.

At the end of each stage of the trial, anthropometric measurements and blood samples were collected from every participant. Throughout the whole period of the study, participants remained under the control of the dietician, who supervised the proper course of nutritional intervention and kept regular telephone contact to check compliance to the protocol, correctness and regularity in the administration of the appropriate oil. All study participants maintained their current lifestyle, including eating habits and physical activity during the whole study period.

### 2.3. Biochemical Analysis

A venous blood sample (7.5 mL) was collected according to standard methods following an overnight fast and was centrifuged immediately. The plasma was collected in a tube and stored at −70 °C until analysis. In all study participants, the following parameters related to the prediction of atherosclerosis risk were assessed: soluble P-selectin (sP-selectin: Human P-selectin/CD62P ELISA Kit, R&D Systems, Minneapolis, MN, USA) and VCAM-1 (sVCAM-1: Human sVCAM-1 Quantikine ELISA Kit; R&D Systems, Minneapolis, MN, USA); ADMA (quantitative ELISA test, BioVendor, R&D Systems, Minneapolis, MN, USA); hsCRP (ELISA method, ELISA Kit Oxis, International Inc., Foster City, CA, USA), lipid profile (total cholesterol (TC), triglycerides (TG), HDL, LDL—enzymatic colorimetric method, Biomerieux Poland, Warsaw, Poland). Additionally, the Atherogenic Index of Plasma (AIP), defined as log(TG/HDL), was calculated for every subject before and after every nutritional intervention.

### 2.4. Statistics

Statistical analysis was performed using STATISTICA 12 software packages (StatSoft Inc., Tulsa, OK, USA). The Shapiro–Wilk test was used to assess the normality distribution of quantitative variables. The changes of values (deltas [Δ]) for anthropometric and biochemical parameters before and after nutritional intervention were calculated. The results are presented as mean ± standard deviation (SD) and medians with interquartile ranges. The statistical significance of differences in deltas of the parameters between the groups (AmO vs. RaO) were determined with the use of the Wilcoxon signed-rank test. The significance level was set at *p* < 0.05.

Assuming the probability of a type-I error at an alpha cut-off level of 5% (0.05), the probability of a type-II error at a beta cut-off level of 20% (0.2), the difference of anticipated means equal to 20%, the expected value of standard deviation equal to 30% of the mean, and the enrolment ratio equal to 1, the sample size was estimated at 35. Anticipating a maximum 20% drop out rate, 44 patients were recruited.

The primary outcome was defined as differences in changes of high-sensitive C-reactive protein (hsCRP) concentration between the AmO and RaO group. The secondary outcomes were established as differences in changes of novel markers of atherosclerosis (ADMA, sP-selectin, vascular adhesion molecule-1 (sVCAM)) and lipid profile between the groups studied. In addition, the anthropometric parameters were assessed and compared between AmO and RaO groups.

This study was designed in accordance with the guidelines from the Declaration of Helsinki, and all procedures involving human beings were approved by the local Bioethics Committee of the Institutional Review Board at the Poznan University of Medical Sciences, Poland (approval number 359/14). Written informed consent was obtained from all participants. The trial was registered in the German Clinical Trials Register (DRKS-ID: DRKS00014046; URL: https://www.germanctr.de/).

The project was awarded a research grant from the NUTRICIA Foundation (number RG2/2017). The idea, design, analysis, and presentation of study results in this article were not influenced by the NUTRICIA Foundation.

## 3. Results

Of the 55 potential participants, 44 entered the study and were randomly assigned for cross-over, including 23 patients from the AmO to the RaO and 21 patients from the RaO to AmO. None of the participants withdrew, and no one was excluded during the course of the study.

### 3.1. Anthropometric Parameters

The differences in changes of values (Δ) for anthropometric parameters between groups are presented in Table 3. There were no significant differences between changes of BW, height, WC, WHR, and BMI between the groups studied.

### 3.2. Biochemical Parameters

#### 3.2.1. hsCRP

There were no significant differences in changes of hsCRP concentration (ΔhsCRP) between AmO and RaO groups, as shown in Table 4.

#### 3.2.2. Novel Markers of AT

There were no significant differences in changes of ADMA, sVCAM, and sP-Selectin (ΔADMA, ΔsVCAM, and ΔsP-Selectin, respectively) between the AmO and RaO groups, as presented in Table 4.

#### 3.2.3. Lipid Profile

The differences in changes of TC, LDL, HDL, triglycerides (TG), and AIP (ΔTC, ΔLDL, ΔHDL, ΔTG and ΔAIP, respectively) between AmO and RaO groups are presented in Table 5. The TC and LDL concentrations increased significantly in AmO group in comparison with RaO, with no significant differences in HDL and TG changes between groups.

## 4. Discussion

Our study was designed to assess the potential antiatherogenic effect of AmO supplementation in a Central European population of obese and overweight subjects. To the best of our knowledge, it is the first trial to analyze the impact of amaranth oil on concentrations of novel markers of AT in comparison with rapeseed oil supplementation.

The strength of this study was its randomized, double-blinded design and analysis of new markers of AT. The cross-over design removed the “patient effect”, thereby reducing variability and increasing the precision of estimation. To diminish the carryover effect, the washout period was incorporated to the study design. All the participants finished both nutritional interventions. The limitation of the study was that more women than men participated. The hypertension was not the exclusion criterion due to its widespread occurrence. Few of the patients took hypotensive drugs during the study. Although the hypotensive drugs may interfere the lipid profile, in our project the use of the drugs remained unchanged during the whole period of the study. Moreover, the study was designed as a cross-over trial to remove potential “patient effect”.

The primary outcome of the study, the differences in changes of hsCRP concentration between the AmO and RaO groups, reflected the inflammatory status, the paramount event in atherogenic plaque formation [11]. According to the American Heart Association, levels over 3 mg/dL signify increased cardiovascular risk. Hs-CRP directly participates in atherogenesis by modulating endothelial function via expression of adhesion molecules, selectins, and MCP-1 [13,16]. In our study, we did not find any significant difference between changes of hsCRP in the AmO and RaO groups. As the amaranth oil contains high amounts of plant lignans, known for their anti-inflammatory properties [17], we expected a decrease in hsCRP in the group supplemented with amaranth oil. To date, there has only been one study describing the impact of amaranth on hsCRP level in the literature. Kabiri et al. reported a decrease in CRP concentration in rabbits fed a cholesterol-rich diet supplemented with extracts of *Amaranthus caudatus* [18]. The discrepancy between the results of Kabiri and our study could be explained by differences in the form (extracts versus oil) and species of the used plant. Furthermore, the study of Kabiri was carried out on animals.

The secondary study outcome focused on the novel markers of AT and lipid status of the participants. This was the first trial to assess the impact of AmO vs. RaO supplementation on adhesion molecules and ADMA concentrations. ADMA is an endogenous inhibitor of endothelial nitric oxide synthase (eNOS), which is thought to be directly implicated in the regulation of the vascular redox state in human AT by affecting superoxide generation and NO bioavailability [19]. As ADMA reflects the endothelial status and adhesion molecules play a key role in atherogenesis process [20,21], we hypothesized that AmO supplementation would affect their concentrations in comparison to RaO. However, the results of our study did not support this hypothesis, but AmO supplementation significantly affected the lipid profile. Interestingly, the TC and LDL increased significantly in the AmO group in comparison with the RaO group, but there were no significant differences in changes of HDL and TG between the two groups. The increase of TC and LDL was unexpected. In the literature, most authors attribute the hypolipemic properties of AmO to high levels of squalene, the substance involved in cholesterol synthesis [22]. Qureshi and co-workers studied the influence of amaranth intake on cholesterogenesis in chickens, showing that serum TC and LDL were lowered by 10%–30% and 7%–70%, respectively, in birds fed amaranth-containing diets [23]. Also, it was observed that hamsters receiving a hypercholesterolemic diet supplemented with amaranth oil had serum levels of TC and non-HDL decreased by 15% and 22%, respectively, in comparison with the control group [24]. Although most studies regarding the effect of AmO on lipid profile were designed using animal models, so far there was one study in humans, performed by Martirosyan and co-workers [25]. They investigated the effects of AmO in patients suffering from coronary heart disease and hypertension with obesity, showing that AmO supplementation decreased significantly TC, TG, LDL, and VLDL. Although the study was a large, randomized placebo-controlled trial, they used a different type of AmO (*Amaranthus hybridus L*. vs. *Amaranthus cruentus*) as well as different dose (6–12–18 mL/per day vs. 20 mL/per day) in comparison with our study. Interestingly, the diet used by Martyrosyan et al. was a lower-energy diet (1600 kcal vs. 2154 kcal) and contained lower amounts of fat (18% vs. 38%), both of which could affect the results. Nonetheless, our results agree with two studies. De Castro and co-workers demonstrated that AmO, and its component squalene, did not exert a hypocholesterolemic effect in hamsters fed a diet containing high amounts of saturated fat and cholesterol [26]. Miettinen et al. reported that dietary RaO supplementation reduced serum LDL cholesterol by 10% from initial values during a 6 week baseline period, but addition of 1 g squalene to rapeseed oil for 9 weeks resulted in increases of serum TC, VLDL, IDL, and LDL concentrations by 12%, 34%, 28%, and 12%, respectively [27]. Furthermore, AmO is rich in palmitic acid (20.4% in AmO vs. 3.6% in RaO), which is associated with an increased risk of coronary heart disease [28]. However, Clandinin et al. indicated that palmitic acid has no effect on serum lipoprotein profiles in the presence of recommended intakes for 18:2n-6 acids [29]. As AIP is a strong marker to predict the risk of AT and coronary heart disease [30,31,32], we assessed the aforementioned parameter. According to the calculated value, patients can be qualified to the group of low, intermediate, or high risk. The baseline AIP mean and median were low, and the differences in changes of AIP between the groups were not significant. The results corresponded with outcomes of HDL and TG changes between the groups.

Finally, the search for genetic determinants of differences in response to nutritional interventions in inhabitants of different regions requires further exploration. In a recent study, Lankinen et al. reported that the inflammatory response to dietary linoleic acid depends on genotype [33]. In our previous study, we compared the impact of Mediterranean diet and Central European diet on the markers of AT, the authors showed the effects of Mediterranean diet do not exceed Central European Diet in a Central European population [34]. In the past, amaranth together with beans and maize were a staple of the native Central American diet [23]. According to climate changes and the need to identify new species that can be grown in drought conditions, amaranth was brought to Europe as a promising substitute crop [35]. However, we assume, it may require an unspecified period of time to adapt the metabolomic and genomic profiles of the population to routine consumption of new nutrients.

## 5. Conclusions

In conclusion, the use of AmO instead of RaO may promote a proatherogenic lipid profile in obese and overweight inhabitants. However, whether this effect is limited to Central European inhabitants remains to be elucidated, therefore, regional differences in response to nutritional interventions should be focus of further research.

## Figures and Tables

**Figure 1 nutrients-11-03069-f001:**
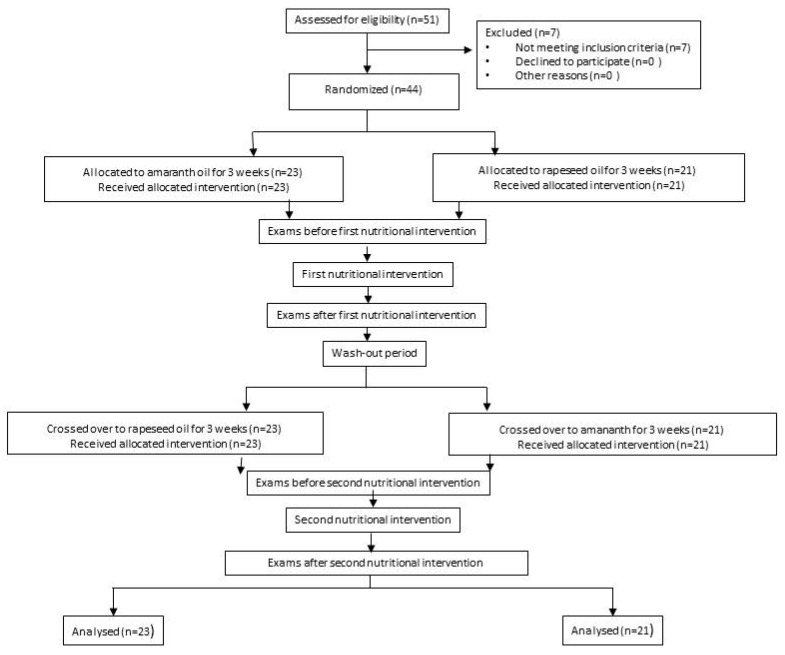
Study flowchart.

**Table 1 nutrients-11-03069-t001:** Baseline characteristics of the study groups.

	Mean ± SD	Median (1st–3rd Quartile)
Age (years)	48.77 ± 10.21	49.00 (42.00; 56.25)
Height (cm)	168.0 ± 8.6	168.3 (160.0; 174.0)
BW (kg)	87.7 ± 15.1	89.8 (73.4; 100.3)
WC (cm)	99.2 ± 13.3	99.0 (90.8;108.0)
WHR	0.89 ± 0.08	0.91 (0.83; 0.95)
BMI (kg/m^2^)	31.00 ± 4.61	30.60 (27.44; 33.16)
hsCRP (mg/dL)	4.59 ± 5.64	2.20 (0.98; 4.90)
ADMA (nmol/L)	1.36 ± 1.15	0.84 (0.64; 1.69)
sVCAM (ng/mL)	851.49 ± 199.47	776.80 (724.43; 974.53)
sP-selectin (ng/mL)	127.66 ± 29.17	125.00 (11.53; 145.42)
TC (mg/dL)	223.02 ± 37.17	219.00 (196.00; 234.50)
LDL (mg/dL)	135.62 ± 37.99	125.65 (113.28; 153.00)
HDL (mg/dL)	56.43 ± 15.87	53.00 (43.00; 66.25)
TG (mg/dL)	150.16 ± 70.08	129.50 (113.25; 189.25)
AIP	0.04 ± 0.28	0.07 (−0.17; 0.22)

BW, body weight; WC, waist circumference; WHR, waist-to-hip ratio; BMI, body mass index; hsCRP, high-sensitive C-reactive protein; ADMA, assymetric dimethylarginine; sVCAM, vascular adhesion molecule-1; sP-selectin, soluble P-selectin; TC, total cholesterol; LDL, low-density cholesterol; HDL, high-density cholesterol; TG, triglycerides; AIP, Atherogenic index of plasma.

**Table 2 nutrients-11-03069-t002:** Diet analysis of enrolled patients.

Analyzed Basic Nutrients	Patients (*n* = 44)	
	Mean ± SD	Median (1st–3rd Quartile)
Energy [kcal]	2153 ± 656	2072 (1642; 6870)
Energy from fats (%)	37.6 ± 6.7	35.7 (32.9; 42.2)
Energy from protein (%)	15.3 ± 2.4	15.4 (13.9; 16.7)
Energy from carbohydrates (%)	47.1 ± 7.2	47.5 (42.2; 52.9)
Protein (g)	80.2 ± 21.3	78.8 (64.6; 93.4)
Fat (g)	89.8 ± 21.3	85.9 (72.5; 102.5)
Cholesterol (mg)	331 ± 128	316 (235; 391)
SFA (g)	28.8 ± 11.9	27.1 (20.6; 35.1)
MUFA (g)	38.7 ± 13.5	37.1 (31.2; 43.3)
PUFA (g)	15.9 ± 4.6	15.1 (12.9; 19.0)
Carbohydrates (g)	262.6 ± 81.6	259.1 (197.3; 312.9)

SFA, saturated fatty acids; MUFA, mono-unsaturated fatty acids; PUFA, polyunsaturated fatty acids.

**Table 3 nutrients-11-03069-t003:** Differences in the changes of values (Δ) for anthropometric parameters between groups supplemented with AmO and RaO.

Δ	AmO		RaO		*p*-Value
	Mean ± SD	Median (1st–3rd Quartile)	Mean ± SD	Median (1st–3rd Quartile)	
BW (kg)	−0.2 ± 1.3	−0.1 (−0.7; 0.5)	0.0 ± 1.6	0.1 (−0.5; 0.8)	0.274
WC (cm)	−0.9 ± 3.7	−1.0 (−4.0; 1.3)	−0.4 ± 3.1	0.0 (−2.0; 2.0)	0.484
WHR	0.00 ± 0.03	0.00 (−0.02; 0.03)	0.00 ± 0.03	0.00 (−0.03; 0.02)	0.882
BMI (kg/m^2^)	0.07 ± 0.47	−0.04 (−0.24; 0.17)	0.01 ± 0.56	0.04 (−0.19; 0.27)	0.232

AmO, amaranth oil; RaO, rapeseed oil; BW, body weight; WC, waist circumference; WHR, waist-to-hip ratio; BMI, body mass index.

**Table 4 nutrients-11-03069-t004:** Differences in the changes of hsCRP concentration (ΔhsCRP) between AmO and RaO groups.

Δ	AmO		RaO		*p*-Value
	Mean ± SD	Median (1st–3rd Quartile)	Mean ± SD	Median (1st–3rd Quartile)	
hsCRP (mg/dL)	0.18 ± 3.03	0.15 (−0.43; 0.75)	−0.32 ± 4.99	0.00 (−0.73; 1.30)	0.554
ADMA (nmol/L)	−0.01 ± 0.34	0.00 (−0.17; 0.10)	0.08 ± 0.41	0.06 (−0.02; 0.25)	0.135
sVCAM (ng/mL)	−9.12 ± 127.94	4.45 (−103.78; 57.58)	−50.83 ± 128.03	−62.95 (−112.73; 7.20)	0.107
sP-selectin (ng/mL)	−2.51 ± 19.05	−3.2 (−17.29; 9.63)	−2.11 ± 19.78	−5.36 (−15.30; 10.62)	0.616

AmO, amaranth oil; RaO, rapeseed oil; hsCRP, high-sensitive C-reactive protein; ADMA, assymetric dimethylarginine; sVCAM, vascular adhesion molecule-1; sP-selectin, soluble P-selectin.

**Table 5 nutrients-11-03069-t005:** Differences in changes of TC, LDL, HDL, and TG (ΔTC, ΔLDL, ΔHDL and ΔTG respectively) between the AmO and RaO groups.

Δ	AmO		RaO		*p*-Value
	Mean ± SD	Median (1st–3rd Quartile)	Mean ± SD	Median (1st–3rd Quartile)	
TC (mg/dL)	5.52 ±35.61	11.00 (−8.25; 26.00)	−8.43 ± 17.65	−5.00 (−20.25; 5.25)	0.002
LDL (mg/dL)	4.43 ± 34.96	11.20 (−8.2; 20.7)	−7.55 ± 16.41	−7.2 (−17.00; 4.03)	0.002
HDL (mg/dL)	0.54 ± 6.26	0.50 (−3.25; 5.00)	−0.65 ± 6.81	0.00 (−5.00; 4.25)	0.805
TG (mg/dL)	7.43 ± 62.96	0.50 (−21.25; 23.25)	1.39 ± 51.66	−1.00 (−28.75; 29.25)	0.788
AIP	0.01 ± 0.15	−0.02 (−0.09;0.08)	0.00 ± 0.21	−0.28 (−0.15; 0.10)	0.427

TC, total cholesterol; LDL, low-density cholesterol; HDL, high-density cholesterol; TG, triglycerides; AIP, Atherogenic Index of Plasma.

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
