# Peer review of "Amaranth Oil Increases Total and LDL Cholesterol Levels without Influencing Early Markers of Atherosclerosis in an Overweight and Obese Population: A Randomized Double-Blind Cross-Over Study in Comparison with Rapeseed Oil Supplementation"

_nutrients, 2019, doi:10.3390/nu11123069_

Round 1

Reviewer 1 Report

REVIEWER COMMENTS ON nutrients- 654987

Comments and suggestions to the authors

General comments: The authors should be careful about the placement of table captions. Where it must be indicated the main explanation, data presentation and all abbreviations (even if they are already described in text). Figure should be redone for a better image quality, and figure caption must be included as in tables.

Specific comments: L68: Did the authors considered, as exclusion or relevant point, the use of pharmacological treatment for obesity or dyslipemia (i.e. statins, etc.), the presence and treatment of arterial hypertension, both systolic and diastolic blood pressure measurements, physical activity index…? If so, the information must be included in this manuscript. If not, they should explain and at least indicate the possible effects of these parameters, among other that were not considered, into the hypothesis and discussion. L94-95: the doses of oil were administrated in exchange of doses of fat. Does this means that fat was considered equal in the frame of all patient’s diet, regardless of the FA composition? Atherogenic Index of Plasma (AIP) is a strong marker to predict the risk of atherosclerosis and coronary heart disease, did the authors calculate this index? If not, please include, correlate and discuss. As in the previous point, glucose levels are an important data that should be included for proper discussion. In this matter, statistical correlations between all parameters must be included and discussed in text. Conclusions must be supported with all the information required.

Author Response

Dear Reviewer,

We would like to thank you for your thoughtful suggestions. Below please find the answer to your comments.

We have made corrections in the table and figure captions as suggested - thank you for your vigilance. The quality of the figure has also been improved.

In reference to the question about possible treatment of obesity and hypertension of the patients- the use of statins was an exclusion criterion (we corrected the list in in Materials & Methods section- line 71) - none of the patients received statins. The hypertension was not the exclusion criterion due to its widespread occurrence. Few of the patients were treated with hypotensive drugs, however the dose and type of the medications were the same before and within the whole study. Moreover, the study was designed as a cross-over trial to remove potential “patient effect”.  According to that, we assumed, it should not influence the results significantly. The aforementioned issue has been posted and discussed in Discussion section (line 193-197).

We replaced 20g of oil (or other fat) by either AmO or RaO (and RaO or AmO in the second phase, respectively) so that the energy value of the diet did not change during the study. The patients were asked not to change the diet during the study. According to dietary analyses, the only difference in FA content was related to the fact of AmO/RaO intake. Moreover, the study was designed as a cross-over trial to remove potential “patient effect”.

We calculated and analyzed Atherogenic Index of Plasma as, indeed, it is a valuable parameter for assessment of the risk of atherosclerosis (Results and Discussion Section –line 245-249).

Thank you very much for your suggestions about the use of glucose levels, systolic and diastolic blood pressure measurements, physical activity index etc. Analysis of physical activity, changes in systolic and diastolic blood pressure and the parameters of carbohydrate metabolism were not the subject of the study. The participants were asked to maintain the physical activity on the same level during the whole period of the study (line 113 Material&Methods section). Moreover, the cross-over design of the study allowed to reduce the risk of potential “patient effect”.

Kind regards,

Monika Dus-Zuchowska

Jaroslaw Walkowiak

Reviewer 2 Report

My concerns:

1. Introduction: Authors need to justify use of RaO as a control oil nutrient supplement.

2. Discussion: Lane 170-Strenght instead of strengthen

Author Response

Dear Reviewer,

Thank you for your opinion about the manuscript. We have made minor corrections as suggested (line 189) - thank you for your vigilance.  Rapeseed oil was administered in the control group as we intended to apply the oil that is commonly used in our region.   

Kind regards,

Monika Dus-Zuchowska

Jarolaw Walkowiak

Reviewer 3 Report

The manuscript describes well designed and executed study where authors investigated anti-atherogenic properties of amaranth and rapeseed oils supplementation in obese or overweight subjects. Authors showed that both supplements were not different by the effect on early markers of atherosclerosis. At the same time amaranth oil statistically significantly increased plasma level of total and LDL cholesterol, this potentially can increase risk of developing atherosclerosis. Rapeseed oil supplementation was accompanied by decreasing the concentration of the same lipids.

One small note: In line 178 of the manuscript authors mentioned the level of CRP as 3 mg/L. I believe it should be 3 mg/dl.

Author Response

Dear Reviewer,

Thank you for your opinion about the manuscript. We have made proper corrections as suggested (line 200) - thank you for your vigilance. 

Kind regards,

Monika Dus-Zuchowska

Jaroslaw Walkowiak